# A Dual-Branch Fusion of a Graph Convolutional Network and a Convolutional Neural Network for Hyperspectral Image Classification

**DOI:** 10.3390/s24144760

**Published:** 2024-07-22

**Authors:** Pan Yang, Xinxin Zhang

**Affiliations:** 1College of Computer and Information Engineering, Xiamen University of Technology, Xiamen 361024, China; yangpan@s.xmut.edu.cn; 2Fujian Key Laboratory of Pattern Recognition and Image Understanding, Xiamen University of Technology, Xiamen 361024, China

**Keywords:** hyperspectral image classification, graph convolution network, convolution neural network, multi-scale superpixel segmentation, multi-fusion information

## Abstract

Semi-supervised graph convolutional networks (SSGCNs) have been proven to be effective in hyperspectral image classification (HSIC). However, limited training data and spectral uncertainty restrict the classification performance, and the computational demands of a graph convolution network (GCN) present challenges for real-time applications. To overcome these issues, a dual-branch fusion of a GCN and convolutional neural network (DFGCN) is proposed for HSIC tasks. The GCN branch uses an adaptive multi-scale superpixel segmentation method to build fusion adjacency matrices at various scales, which improves the graph convolution efficiency and node representations. Additionally, a spectral feature enhancement module (SFEM) enhances the transmission of crucial channel information between the two graph convolutions. Meanwhile, the CNN branch uses a convolutional network with an attention mechanism to focus on detailed features of local areas. By combining the multi-scale superpixel features from the GCN branch and the local pixel features from the CNN branch, this method leverages complementary features to fully learn rich spatial–spectral information. Our experimental results demonstrate that the proposed method outperforms existing advanced approaches in terms of classification efficiency and accuracy across three benchmark data sets.

## 1. Introduction

A hyperspectral image (HSI) offers more refined spectral information than other remote sensing images (e.g., optical and multispectral images), making it advantageous for hyperspectral image classification (HSIC) tasks. HSIC has various applications in military detection, mineral exploration, agricultural production, urban planning, and environmental monitoring [1,2,3,4,5]. However, HSI has inherent characteristics such as high-dimensional data, a limited training sample size, and spectral uncertainty. Predicting the true class of each pixel with high accuracy based on spatial–spectral features in HSI remains a challenge [3] and merits further research.

During early HSIC research, most studies focused on investigating how spectral characteristics functioned in classification, proposing many traditional pixel-based classification methods, covering support vector machines (SVMs) [6], K-nearest neighbors [7], polynomial logistic regression [8,9], etc. The main disadvantage of these methods is that the feature engineering step is a time-consuming task, and the classification accuracy tends to be reduced when highly relevant features or features with low information content are extracted in this step. Given the inherent nonlinear relationship between spectral information and the corresponding materials in HSI, it is challenging to accurately classify such data using traditional machine learning methods.

Deep learning (DL) is regarded as a powerful tool that can be used to solve nonlinear problems. It has been utilized extensively in several image processing tasks, including image classification [10,11], target detection [12], natural language processing [13], etc. Inspired by the success of these applications, HSIC has also benefited from DL, demonstrating promising results. Currently, one of the most important technical indicators of the classification model’s quality is how to avoid the dimension disaster problem and extract useful and discriminative feature information from the high-dimensional feature space of HSIs. In [14,15,16,17,18], most scholars employ 2D CNNs to extract the spatial characteristics in the hyperspectral pixel neighborhood after first performing a principal component analysis (PCA) on the entire set of hyperspectral data to lower the dimensionality of the original space. This method can efficiently extract spatial information and lower computing costs by integrating PCA and CNNs. Nevertheless, spectral information is unavoidably lost during the dimensionality reduction process, which might impact the model’s ability to comprehend input features and its overall classification performance. Li et al. proposed a three-dimensional convolutional neural network (3-D-CNN) that can synchronously analyze spatial spectrum characteristics and produce impressive classification performance [19]. It is worth noting that a diverse-region CNN (DR-CNN) utilized various neighboring areas of the center target pixel; nevertheless, when the label sample size was insufficient, the model’s performance was a cause for concern in terms of generality [20].

In addition, as the data volume increases, the issue of scarce labeling samples worsens, making it challenging to use the above-mentioned CNN-based supervised learning approach in the absence of a sufficient number of labeled training samples. Consequently, scholars began focusing on the semi-supervised learning (SSL) approach, which utilized the labeled data and information from unlabeled samples as a supplement. For instance, Zhou et al. suggested a label propagation approach that makes use of labeled samples to execute label propagation on the full HSI to obtain labels for unlabeled samples [21]. However, because of the sensitivity of the parameters, this approach is prone to noise. The semi-supervised support vector machine (S3VM) included the construction of a support vector machine classifier by combining existing labeled samples and adding a certain proportion of unlabeled samples [22]. Although it produces good classification results, a good generalization performance necessitates careful parameter adjustment. Makhzsani et al. provided a semi-supervised classification method by restricting the reconstruction error and reconstructing HSIs using autoencoders [23]. Common algorithms include sparse autoencoders and variational autoencoders. Good classification results have been obtained using this strategy. However, it is too time-consuming.

Notably, semi-supervised graph convolutional networks (SSGCNs) have demonstrated notable efficacy as one of the most efficient SSL techniques. By effectively processing the local spatial characteristics and global semantic characteristics in HSIs, the utilization of features from unlabeled nodes and the comprehensive learning of the interaction and feature transfer between nodes have led to a notable increase in classification accuracy. However, a traditional GCN can compile and convert features from every graph node’s neighbor, but it only utilizes spectral features and overlooks the significant space structures that are embedded in the original HSI data [24]. Moreover, when dealing with a large number of pixels, the construction and computing costs associated with the graph structure are unfeasible. Compared with the original GCN, Qin et al. proposed a spectral-space GCN (S^2^GCN) that achieved superior accuracy in classification [25]. Nevertheless, this approach only employs a fixed neighborhood size and graph throughout the graph convolution process, making it unable to flexibly capture spectral–spatial information from various local areas and accurately portray the intrinsic relationship between pixels. Consequently, Wan et al. [26] suggested a multi-scale dynamic GCN (MDGCN) that included superpixels in multi-hop graph learning, saving training time while reducing computing complexity. However, integrating multi-scale spatial information utilizing a spatial multi-hop graph structure may lead to classifier deviation, which would impact the classification performance. Li et al. [27] proposed a novel framework called SGML, which can better capture the similarities and differences between samples to improve classification efficiency by combining graph-embedding technology and metric learning methods. However, this network adopts a multi-scale superpixel segmentation technology to process hyperspectral images, which is likely to ignore the pixel features of local details. Dong et al. [28] proposed a weighted feature fusion method combining a convolutional neural network (CNN) and graph attention network (GAT), which led to the proposal of a new solution for dual-branch fusion networks, but the classification effect still needs to be improved.

Hence, in this paper, we propose a dual-branch fusion of a GCN and CNN, namely DFGCN, to achieve superior hyperspectral image classification outcomes. First, a multi-scale superpixel segmentation method is employed in the GCN branch to optimize the utilization of various feature information points related to shapes and sizes. Additionally, this approach significantly reduces computational cost by converting the algorithmic calculation unit from individual pixels to superpixels. Next, fusion-adjacent matrices are created based on superpixels for each scale to better measure the similarity between the graph nodes, resulting in more efficient graph convolution and stronger node representation. Then, a spectral feature enhancement module between the two graph convolutions enhances the most important channels of information during data transmission. In the CNN branch, we designed a convolutional network with an attention mechanism to concentrate on extracting detailed features of local areas. Through the fusion of the multi-scale superpixel features from the GCN branch and the local pixel features from the CNN branch, our proposed approach comprehensively captures and fully learns rich spatial–spectral information, thereby enhancing classification performance. The following are the novel aspects of this study: (1)The methodology adopted in this research involves the construction of a fusion adjacency matrix following the segmentation of an HSI using multi-scale superpixel segmentation. The incorporation of the Pearson correlation coefficient as a supplement to the construction of a similarity function based on Euclidean distance is a critical aspect of this study, and the weight ratio between the two is of paramount importance. The introduction of new adjacency matrices plays a vital role in discovering novel graph structures, facilitating the learning of more powerful node representations and enhancing the effectiveness of the graph convolutions. The proposed technique enables the extraction of spatial information features that are more comprehensive and discriminative than existing methods do.(2)The spectral feature enhancement module was designed in the middle of the two graph convolutions to enhance important channel information in a self-supervised way to extract more important spectral information.(3)We fused the GCN branch based on multi-scale superpixel segmentation with the CNN branch, which included an attention mechanism, to fully extract the long-distance contextual information and local detail features of the HSI. Furthermore, our extensive experiments demonstrated that the proposed DFGCN outperforms several widely used and advanced classification techniques in terms of classification results.

The remainder of this article is arranged as follows: Our methods are introduced in Section 2 and include the entire architecture, a synopsis of the superpixel segmentation, and a detailed implementation of the proposed DFGCN. Our experimental data sets and evaluation indicators are described in Section 3. Our extensive experiment results are presented in Section 4. Further analysis and discussion are included in Section 5. Section 6 presents our conclusions.

## 2. Methods

### 2.1. Architecture of the Proposed DFGCN

This section describes the proposed DFGCN, which can be seen in Figure 1. It is primarily divided into two branches: the GCN branch, based on multi-scale superpixel segmentation, and the CNN branch with an attention mechanism. In the GCN branch, we perform adaptive multi-scale superpixel segmentation on the first principal component after reducing the dimensionality of the HSI using the PCA method. For each scale, we map the graph nodes from the pixel scale to the superpixel scale. Then, we carry out fusion adjacency matrix construction (FAMC). We establish a spectral feature-enhanced module between the two graph convolutions and employ them for spectral feature extraction. In the CNN branch, we design a convolutional network with an attention mechanism to focus on extracting detailed features of local areas. Finally, we fuse the complementary features of the two branches and send them to the classifier. In the following section, we will provide a detailed description of the main DFGCN implementation procedures, including the multi-scale superpixel segmentation method, construction of the fusion adjacency matrix, design of the spectral feature improvement module, and structure of the CNN branch.

### 2.2. Superpixel Segmentation

Superpixel segmentation is a technique that enhances the ability to extract semantic information from images by aggregating pixels in the image that have similar color and texture features into a more significant and recognizable portion [29]. This new portion serves as the fundamental component of subsequent image processing, which can greatly reduce the computational burden, as seen in Figure 2 below. Furthermore, superpixel segmentation has already been employed as a preprocessing technique in many HSIC methods and has proven to be effective [30]. For example, Li et al. proposed a symmetric graph metric learning framework based on a multi-scale adaptive superpixel segmentation technique to increase classification efficiency using the graph’s structural characteristics and metric learning technology [27]. Jia et al. presented methods for clustering pixels with similar spectral characteristics by carrying out weighted label propagation on superpixels [31]. They reduced the computing time and obtained a notable classification performance. Specifically, entropy rate segmentation (ERS) is usually chosen to produce superpixels because of its efficacy compared with other methods [32]. In summary, ERS could be translated as the solution to the following objective function, since it is a graph-based technique:(1)$$\underset{p}{\arg}\;\max E(P)+\gamma A(P)$$

Here, E(P) is the limitation of the entropy rate, which is used to create homogeneous clusters. A(P) is a balanced constraint that lowers the amount of imbalanced superpixels by requiring clusters to have comparable spatial sizes. γ represents the balance of the constraint’s weight coefficient, which must be greater than or equal to 0.

### 2.3. Fusion Adjacency Matrix Construction

Next, we will introduce how to build a fusion-graph-adjacent matrix using an HSI after multi-scale superpixel segmentation. First of all, we will briefly introduce the process of converting pixels into superpixels. Because superpixels may automatically modify their size and shape based on the HSI content, they are an excellent way to describe land cover. Consequently, we leverage superpixels to make further graph learning easier. In this case, the superpixel’s value is determined by weighing the average of the pixels that make up one superpixel: $V_{sp}=\{v_{1},v_{2},\ldots ,v_{J}\}$.

Let Xlsp∈ℝS1×L={xl1,xl2,…,xlS1} and Xulsp∈ℝS2×L={xul1,xul2,…,xulS2} be the labeled and unlabeled superpixels, respectively. The length of every superpixel is denoted by L. Superpixels with labeled and unlabeled samples are denoted by S1 and S2, respectively, and S1+S2=S. By means of majority voting of the contained pixels, the corresponding labels of Xlsp are selected. After that, every superpixel is utilized to create the graph G=(X,A), with X∈ℝS×L representing each graph node’s superpixel characteristic.

The majority of currently existing GCN-based HSIC methods employ a single Euclidean distance to build the similarity function of the graph adjacency matrix [27,33]. As an intuitive distance measurement method, Euclidean distance can measure the degree of difference between pixels, expressed by the geometric distance between hyperspectral pixels, more effectively (Formula (2)). However, it does not properly account for the linear correlation between data, which could lead to a graph adjacency matrix that may not accurately capture the complex data features in the HSI, thereby impacting the accuracy and stability of classification and weakening the robustness of the algorithm.
(2)$$d(x,y)=\sqrt{\sum_{i}^{n}{\left(x_{i}-y_{i}\right)}^{2}}$$

Therefore, we consider introducing the Pearson correlation coefficient as a supplement to construct the similarity function based on the Euclidean distance. The Pearson correlation coefficient measures the similarity between variables based on their covariance (Formula (3)), which takes into account the linear relationship between variables to determine whether variables change in similar or opposite trends.
(3)$$p(x,y)=\frac{\sum_{i=1}^{n}(x_{i}-\bar{X})(y_{i}-\bar{Y})}{\sqrt{\sum_{i=1}^{n}{\left(x_{i}-\bar{X}\right)}^{2}}\sqrt{\sum_{i=1}^{n}{\left(y_{i}-\bar{Y}\right)}^{2}}}$$

Constructing a similarity function, which more effectively captures the spatial relationship, feature similarity, and correlation between different pixels in an HSI, is achieved by combining the Euclidean distance and Pearson correlation coefficient. This creates more comprehensive feature information while improving the performance of the HSIC tasks and enhances the classification efficacy and accuracy. The structural function of the graph-adjacent matrix is shown in Formula (4), where adj denotes that two graph nodes are adjacent and α represents the weight ratio of two similar measurement methods.
(4)$$A_{i,j}=\left\{\begin{array}{cc} \alpha p(x^{i},y^{j})+(1-\alpha )e^{-\beta d(x^{i},y^{j})} & if:\;x^{i}\;adj\;x^{j}\\ 0, & else \end{array}\right.$$

### 2.4. Graph Convolutional Network

When it comes to spectral-based convolutional graph neural networks, the GCN is one of the most often used techniques. Its main use in topological graphs is to extract pertinent vertices and edges’ spatial characteristics [34,35]. Notably, Kipf and Welling [24] developed an efficient layer-by-layer propagation method that can encode node properties and the local graph structure, leading to a more stable state, using Chebyshev polynomials for estimating the convolution kernel. In short, from the Fourier perspective of graph Laplacian [36], the definition of the convolution operation is as follows:(5)$$g_{\theta }*x_{g}=Ug_{\theta }U^{T}x_{g}$$

Among them, the convolution filter parameterized by θ is represented by $g_{\theta }$, while $x_{g}$ represents the graph signal. The eigenvector matrix of the normalized graph Laplacian is represented by the symbol U, which may be written as $L=I-D^{-(1/2)}AD^{-(1/2)}=UU^{T}$. The identity matrix with the proper size is represented by the symbol I. The graph’s degree matrix is denoted by D, while the adjacent matrix is represented by A. The diagonal matrix Λ corresponds to the eigenvalues of L. Next, the authors in [34] used the truncated translational Chebyshev polynomial $T_{k}(x)$ to approximate $g_{\theta }$. This can be stated as follows:(6)$$g_{\theta }*x_{g}\approx \sum_{k=0}^{K}\theta_{k}T_{k}(\hat{L})x_{g}$$
where $\theta_{k}$ is the kth Chebyshev coefficient, and shifted, $\hat{L}=(2/\lambda_{\max})L-I$. $\lambda_{\max}$ is the greatest eigenvalue of L. Notably, this operation is K-localized, since it uses the Kth-order polynomial of the Laplacian. The layer-by-layer convolution process is further approximated and restricted to K=1 using the GCN [24]. The computation formula is as follows:(7)$$g_{\theta^{\prime }}*x_{g}\approx \theta_{0}^{\prime }x_{g}+\theta_{1}^{\prime }(L-I)x_{g}=\theta_{0}^{\prime }x_{g}-\theta_{1}^{\prime }D^{-(1/2)}AD^{-(1/2)}x_{g}$$
where $\theta_{0}^{\prime }$ and $\theta_{1}^{\prime }$ are shared by the two free parameters across the entire graph. Under restrictive conditions of $\theta =\theta_{0}^{\prime }=-\theta_{1}^{\prime }$, (4) can be simplified to
(8)$$g_{\theta }*x_{g}\approx \theta (I+D^{-(1/2)}AD^{-(1/2)})x_{g}$$

To prevent the issue of disappearance/explosion gradients and numerical instability, I+D−(1/2)AD−(1/2)→D^−(1/2)A^D^−(1/2) performs renormalization in conjunction with A^=A+I and Dij=∑jA^ij. Lastly, the graph convolution can be expressed as follows for the signal X∈ℝN×C (N nodes):(9)Y=D^−(1/2)A^D^−(1/2)XΘ
where Θ∈ℝC×F and F are the trainable convolutional variables and the number of kernels, respectively. The graph convolution’s output is represented by Y∈ℝN×F.

Moreover, considering the highly nonlinear geometric nature of an HSI in the characteristics area, which is susceptible to changes in lighting, environment, atmosphere, time conditions, etc., we may potentially improve the robustness of the experiment by working on the graph [37]. Several studies have used a GCN to classify an HSI, thereby achieving encouraging results. In this work, we further explore how to fully utilize the advantages of graph convolution by supplementing the spatial information across scales and taking the similarities and correlations across nodes into account. Specifically, we spread the labeled sample feature into the unlabeled samples using graph convolution and design, designing the spectral feature enhancement module to study the local correlation of spectral features within nodes. Through multi-scale interaction and deep feature mining operations, we obtain more representative and discriminative features and achieve highly accurate classification results.

### 2.5. Spectral Feature Enhancement Module

According to the GCN theory, the primary purpose of graph convolution is to propagate information across nodes without taking into account how important the internal relationships of nodes are. On the other hand, local and non-local spectral features are highly significant for classification tasks when processing an HSI, as they are tightly associated with nodes in the graph. Thus, we sandwich the spectral feature enhancement module (SFEM) between two graph convolutions. The purpose of this module is to enhance the expressive ability of spectral features so that it can more effectively discern distinctions between various categories. Figure 3 depicts the design of this module. 

To better capture details and heterogeneous information, we initially perform two lightweight one-dimensional convolution(1-D-Conv) operations on the spectral features, first increasing the dimension and then reducing it. The output is then scaled to within the range of 0 to 1 as significant factors of various channels of graph nodes using the sigmoid function. Features of significant channels are then highlighted by performing an element-wise multiplication of these factors with the input graph node features. We also add the original features with the aforementioned outcomes to prevent unnecessary information loss. In using this self-supervision approach, fewer significant spectral features are comparatively limited, while the critical feature expression of channels is boosted. The following formula represents the SFEM:(10)$$h_{2}=h_{1}⊙\left(\sigma \left[w_{2}\sigma \left(w_{1}*h_{1}\right)\right]+1\right)$$
where $h_{1}$ represents the input graph node feature, and $w_{1}$ and $w_{2}$ are the weights of two 1-D-Conv, respectively. ⊙ represents the pixel product operation.

### 2.6. Structure of CNN Branch

Superpixel segmentation technology aggregates pixels in an HSI and represents them with the same features, but the same-spectrum and different-spectrum characteristics of HSI data may lead to erroneous superpixel segmentation, thus affecting the subsequent classification accuracy. Secondly, after treating each superpixel as a graph node, information can only be propagated between each superpixel, ignoring the local spatial spectrum information within the superpixel. Considering the above factors, we designed a CNN branch with an attention mechanism to obtain local detail features to solve the problems of edge smoothing and detail loss during classification, which may be caused by the superpixel segmentation technology. This branch consists of two Squeeze-and-Excitation (SE) attention mechanisms and depthwise separable convolutions. Among them, the SE attention mechanism ensures better classification results with a small amount of calculation. Depthwise separable convolution is a special convolution operation in a CNN that aims to reduce the number of parameters and calculations of the model while improving its efficiency and performance.

The SE module includes three key steps: First, compress the spatial dimension of the input features from a three-dimensional tensor of H × W × C to a tensor of 1 × 1 × C. Second, generate each feature through a fully connected layer. The excitation weight of the channel is used to characterize its importance. Finally, multiply these weights by the original feature tensor to adjust the importance of each channel in the feature map, highlight important features, and suppress the influence of unimportant features, thereby achieving adaptation attention weighting. A structural diagram of this process is shown in Figure 4.

Depthwise separable convolution consists of two steps: depth convolution and pointwise convolution. First, a convolution kernel of size K × K × 1 is applied to each channel of the image of an input size of H × W × B for convolution operation, which will produce B feature maps of sizes of H × W, where each feature map corresponds to a channel of the input image. Then, use a convolution kernel of a size of 1 × 1 × B to perform a point-by-point convolution operation on the feature map obtained via depth convolution, which is equivalent to a linear combination between channels and will eventually produce an output result of H × W × M. Here, M is the number of output channels in the pointwise convolution operation. Compared with traditional convolution methods, depth-separable convolution can reduce the computational cost to 1/K^2^.

## 3. Experiments

We conducted extensive tests using three standard HSI data sets, namely Indian Pines (IndianP), Pavia University (PaviaU), and Kennedy Space Center (KSC), to assess the efficacy of the proposed DFGCN. For comparison, we selected several prevalent and advanced machine learning-, CNN-, and GCN-based methods, including RBF-SVM [6], 2-D-CNN [17], 3-D-CNN [19], GCN [24], S^2^GCN [25], MDGCN [26], and SGML [27]. An Intel (R)Core (TM) i7-8700K CPU with 12 GB of RAM, an NVIDIA RTX3080Ti graphics card, and the TensorFlow deep learning framework were used with all algorithms. This device is manufactured by Intel and located in Santa Clara, CA, USA.

### 3.1. Experimental Data Sets

1.Indian Pines: The Airborne Visible/Infrared Imaging Spectrometer (AVIRIS) above Northwestern Indiana took the image of the IndianP. Its spatial size is 145 × 145, with a resolution of 20 m. It captured reflections of 220 bands, spanning from 0.4 to 2.5 μm in the spectral view. Two hundred bands remain after the removal of the noise and water absorption bands, including sixteen label types. Figure 5a,b depict the false-color composite image and ground truth, respectively. Table 1 contains a detailed list of the training, test, and total sample counts for each class.

2.Pavia University: The Reflective Optical System Imaging Spectrometer (ROSIS) facility in urban settings provided the PaviaU scenario. It consists of 610 × 430 pixels, with a spatial resolution of 1.3 m. After discarding noise bands on the spectral dimension, 103 bands remain, with wavelengths from 0.43 to 0.86 μm, which includes nine label categories of objects. Figure 6a,b depict the false-color composite image and ground truth, respectively. Specific data on the training, test, and total samples of each class are listed in Table 2.

3.Kennedy Space Center: The KSC scene was collected using the AVIRIS sensor above the Kennedy Space Center in Florida and covers 224 bands with wavelengths from 0.4 to 2.5 μm. The KSC maintains 176 bands after the abandonment of low-SNR and suction channels. The space size is 614 × 512 pixels, and the space resolution is 18 m, covering 13 different types of earth objects. The false-color composite image and ground truth are displayed in Figure 7. Specific data on the training, test, and total samples of each class are listed in Table 3.

### 3.2. Evaluation Parameters

We employ widely used assessment parameters, such as the overall accuracy (OA), average accuracy (AA), Kappa coefficient, and F1-Score, to objectively and comprehensively assess the effectiveness of the proposed DFGCN in classification tasks. The OA is the proportion of the accurately classified sample to the entire classification sample, the AA is defined as the average of the classification accurateness, and the Kappa statistic [38] is the difference between the chance of classification results being consistent with the chances of matching actual results, i.e., confusing the line and column of the matrix and between them. It should be noted that Kappa values are between −1 and 1, and better classification models mean that the Kappa value is inclined to 1. The F1-Score is used to measure the performance of a classification model. It takes both the precision and recall of a classification model into account. The closer the F1-Score value is to 1, the better the classification performance is. The following formula represents these parameters:(11)$$\mathrm{OA}=\frac{N_{c}}{N_{a}}$$
(12)$$\mathrm{AA}=\frac{1}{C}\sum_{i=1}^{C}\frac{N_{c}^{i}}{N_{a}^{i}}$$
where $N_{c}$ and $N_{a}$ represent the number of samples that are correctly classified and the overall number of samples, and $N_{c}^{i}$ and $N_{a}^{i}$ coincide with the sums of each category in $N_{c}$ and $N_{a}$, respectively.
(13)$$\mathrm{Kappa}=\frac{\mathrm{OA}-P_{e}}{1-P_{e}}$$

$P_{e}$ is the hypothetical probability of coincidence in chance. The following formula can be used to obtain $P_{e}$:(14)$$P_{e}=\frac{N_{r}^{1}\times N_{p}^{1}+\cdots N_{r}^{i}\times N_{p}^{i}+\cdots +N_{r}^{C}\times N_{p}^{C}}{N_{a}\times N_{a}}$$

In the formula, $N_{r}^{i}$ and $N_{p}^{i}$ denote the number of actual samples for every class and the number of predicted samples in every class, respectively.

## 4. Experimental Results

### 4.1. Experimental Settings

In this section, we conduct intricate tests to maximize the usefulness of the suggested DFGCN. First of all, we select three levels of superpixel segmentation, corresponding to the subsequent three different spatial scales. In the process of constructing the adjacent matrix, we use the combination of the Euclidean distance and Pearson correlation coefficient to construct the similarity function. We create a weight ratio between the two that ranges from 0.1 to 0.9 and run tests in increments of 0.1. As shown in Figure 8, we discover that the weight parameter value of 0.5 produces the best experimental outcomes on the three benchmark HSIC data sets. Thus, we set it to 0.5 in our ensuing studies. 

In addition, we analyze the impact of the parameters in the SFEM on the model’s performance. Specifically, the parameter varies in the range of {4, 8, 16, 32}. Figure 9 displays the overall categorization accuracies. It can be seen that, for the IndianP, PaviaU, and KSC data sets, respectively, the best classification results are obtained when γ equals 8, 8, and 32. 

Finally, in the CNN branch, the convolution kernel size is 3 × 3, the first convolution layer contains 128 convolution kernels, and the second layer contains 64 convolution kernels. We use the full-batch gradient descent method and the effective Adam approach for parameter optimization. The KSC data set has a learning rate of 5 × 10^−5^, whereas the IndianP and KSC data sets are both set at 5 × 10^−4^. Furthermore, 500 is the stated epoch number.

### 4.2. Classification Performance

To illustrate the advantages of the suggested method, we contrast DFGCN with the currently popular advanced HSIC approaches, where the setting of the parameters aligns with the relevant paper. Specifically, the RBF-SVM [6] uses the RFB kernel to describe an SVM. Among the popular CNN-based methods are 2-D-CNN [17] and 3-D-CNN [19], which are widely cited as benchmarks for spectral–spatial HSIC. In addition, the GCN [24] introduced the DL paradigm into graph learning. S^2^GCN [25] further utilizes the spectral and spatial features in an HSI. Using multi-scale spatial information, the MDGCN [26] is one of the most recently proposed SOTA approaches. SGML [27] is a newly proposed symmetric graph metric learning framework. Table 4, Table 5 and Table 6 provide the details of the mean per-class accuracy, AA, OA, Kappa, and F1-Score values for the above-mentioned models. Bold text indicates the rows with the highest accuracy levels.

In Table 4, we observe that the proposed DFGCN acquires the best results for the AA, OA, and Kappa. As predicted, DL-based techniques outperform conventional RBF-SVM methods. Regarding the AA, the semi-supervised learning approach based on a GCN outperforms the supervised learning approach based on a CNN, which, to some degree, illustrates the potential of graph convolution in HSIC. More significantly, the accuracies of most categories in the DFGCN rank first among these methods, especially for the tenth category, namely Soybean—no-till, for which DFGCN can obtain an accuracy of up to 99.46%, while other approaches find it challenging to identify. This infers that the multi-scale and fusion graphs have a strong capacity to analyze the rich spectral and spatial characteristics contained in an HSI. Moreover, the SFEM can produce more differential discriminative features, thereby improving the performance of the classification.

Table 5 displays the classification performance of several algorithms using the PaviaU data sets. Given that the training samples of the data set are relatively dispersed, most approaches find it challenging to achieve accurate classification, and, therefore, the classification accuracy is relatively low. In terms of a comprehensive consideration of spectral and spatial information, 3-D-CNN and S^2^GCN achieve improved performance compared with 2-D-CNN and GCN, which confirms the importance of spectral and spatial feature merging in HSIs. Most importantly, we find that the DFGCN also achieves an obvious advantage over the MDGCN. The AA, OA, Kappa, and F1-Score received an inspiring 2.72%, 2.32%, 2.98%, and 2.28% increase, respectively. Compared with SGML, the DFGCN has a better classification effect at image edges, which may be related to the local detail features extracted by the CNN branch. It makes sense to assume that the suggested framework can extract spectral and spatial characteristics more effectively.

From Table 6, we find that the classification results of the DFGCN reach the highest level compared with other classification methods using four parameters, the AA, OA, Kappa, and F1-Score, with 100% correct predictions for up to 11 covered categories. It is worth noting that due to the superpixel segmentation preprocessing, as well as the multi-scale feature fusion, the MDGCN and SGML obtain amazing classification results with fewer parameters, which may be due to the important role of the multi-scale architecture in HSIC. This is another significant justification for this paper’s usage of a multi-scale design. It is evident from the classification performance on three benchmark data sets that a network consisting of a fusion adjacency matrix, SFEM, and multi-scale architecture has obvious benefits.

Regarding qualitative analysis, Figure 10, Figure 11 and Figure 12 plot the classification maps along with the ground truth obtained on different models using the IndianP, PaviaU, and KSC data sets. We discover that the suggested DFGCN performs the best since its classification maps most closely resemble the real-world scenario and have the lowest classification error. Comparatively, the 2-D-CNN only considers spatial features, and the classification effect is not ideal, with large pieces of pepper–salt noise on the classified map. The 3-D-CNN takes spatially adjacent samples into account, obtaining relatively tight classification maps with significantly better classification effects. In addition, the GCN maps also contain some of the dispersed pepper–salt noise, which is associated with the use of spectral features only. Specifically, we note that the proposed DFGCN, in contrast to the MDGCN and SGML, can yield more accurate findings at the regional margins, which are primarily the locations of border samples that are hard to differentiate. This suggests that the proposed DFGCN may be able to identify various surface coverings with similar spectral–spatial information and extract comprehensive and distinctive features.

In addition, we use the PU data set to calculate the ROC curve and AUC value of each category and the macro-average AUC and micro-average AUC for evaluating the performance of the model. The ROC curve shows the trade-off between the true positive rate and the false positive rate under different thresholds. The AUC value represents the area under the ROC curve and is an important indicator to measure the performance of the classifier. The closer the value is to 1, the better the performance is. Figure 13 shows the ROC curve and AUC value of the PU data set. It can be seen that the AUC values of categories 3, 5, 7, 8, and 9 are all 1, and the ROC curve is almost completely in the upper-left corner, achieving a very ideal classification situation. The AUC values of categories 1, 2, 4, and 6 are 0.97, 0.90, 0.97, and 0.98, respectively, which means that the classification ability of the DFGCN in these categories is slightly weaker than that of other categories. Although it is slightly lower than 1, it achieves satisfactory classification results. At the same time, the macro-average and micro-average AUC reached 0.98 and 0.94, respectively, indicating that the DFGCN has a good generalization ability and classification accuracy in multi-category classification tasks.

## 5. Discussion

### 5.1. Ablation Network

To comprehensively evaluate the DFGCN network proposed in this paper, we conducted the following ablation experiments: First, we explored the role of two branches in the DFGCN network, namely the graph convolution branch based on multi-scale superpixel segmentation and the CNN branch in the DFGCN network. Secondly, the role of the SFEM in the GCN branch was evaluated. The experimental results are shown in Table 7. We found that, if the DFGCN lacks any of its branches, its overall classification accuracy is affected, and the experimental effect of the GCN branch is better than that of the CNN branch. 

In addition, as shown in Table 8, we found that when the SFEM is not added, the classification results will decrease slightly, which further verifies that the SFEM can enhance important channel information while transmitting information, thereby improving the classification effect. These experimental results fully demonstrate the effectiveness and robustness of the DFGCN network in hyperspectral image classification tasks.

Furthermore, we apply grayscale images on the PaviaU data set to visualize the importance of different channels at different scales, as shown in Figure 14 below. (a), (b), and (c) represent three different scales. Among them, the vertical axis represents the number of superpixels, and the horizontal axis represents the channel importance of a single superpixel. White represents the strongest importance, and black represents the opposite. In the red box areas in Subfigures (a), (b), and (c), the importance of these superpixels on all channels is surprisingly consistent, which shows that the superpixels in this range are very likely to be in the same category. In addition, in the blue box in Subfigure (a), the orange box in Subfigure (b), and the green box in Subfigure (c), the importance of all superpixels on these channels is almost the same. This shows that these channels have the same importance for the hyperspectral image classification task: some are important, and some are unimportant. Figure 7 also supports the claim that the model has increased its precision to different degrees after the SFEM has been added to the three base data sets, further demonstrating the significant effectiveness of the SFEM in the HSIC task. 

### 5.2. Qualitative Viewpoint Regarding Feature Discrimination

This section shows the fused output of our model’s final graph convolution and the original spectral feature using t-SNE technology. Specifically, we complete the visualization based on t-SNE algorithms using the manifold module in the sklearn package. The t-SNE method requires a feature matrix, which comprises every non-background pixel with spectrum curves serving as feature vectors in the original image. Furthermore, we store the fused output characteristics in the final convolutional layer of the graph as feature vectors. Figure 15 displays the visualization outcomes for IndianP, PaviaU, and KSC. It can be observed that these samples are intermingled in the original spectral domain, making it difficult to classify them into the appropriate category. This situation is substantially better with our approach, particularly in the IndianP and PaviaU data sets. For instance, in Figure 13, the original features of categories 2, 3, 10, and 11 in (a) are haphazardly combined and overlapped in the spectral domain. But in (d), these categories are discernibly distinguished and exhibit an aggregated state. Furthermore, the original features of categories 3, 5, and 6 in (b) are scattered throughout the spectral domain, but these categories start to assemble and separate from other categories after our approach is applied. The aforementioned visualization outcomes illustrate that the suggested DFGCN can enhance our feature classification ability and achieve better classification results.

### 5.3. Computational Cost

The computational cost of the suggested DFGCN in comparison to the other graph-based baseline techniques, namely GCN, S^2^GCN, and MDGCN, is displayed in Table 9. We observe that the computational costs of the GCN and S^2^GCN are greater than those of the MDGCN and DFGCN due to the lack of a preprocessing approach for superpixel segmentation. As a SOTA method, the MDGCN minimizes data volumes due to ultra-pixel technology, significantly reducing the training time. Notably, on the three benchmarks, the proposed DFGCN outperforms the MDGCN by approximately 12 times, 5 times, and 2 times, respectively. Compared with SGML, the DFGCN achieves a better classification performance at the expense of a certain efficiency, which is acceptable. The DFGCN can achieve a better classification performance with less training time, significantly alleviating the pressure of real-time needs in the HSIC task. This further demonstrates that the framework we have suggested is a valid and effective HSIC model. 

## 6. Conclusions

This study presents a dual-branch fusion of a GCN and convolutional neural network for HSIC, named the DFGCN. In the GCN branch, we initially segmented an HIS using a multi-scale superpixel segmentation method and constructed a fusion adjacency matrix for each scale. Based on the Euclidean distance, we added the Pearson correlation coefficient as a supplement to the feature correlation to better measure the similarity between nodes and extract more comprehensive and discriminative spatial features when building the adjacency matrix. This allowed us to find new graph structures to perform more effective graph convolution and learn more powerful node representation. Moreover, the spectral feature enhancement module was designed between two graph convolutions. In using self-supervision, this module can enhance the feature expression of significant channels while comparatively limiting irrelevant spectral characteristics. In the CNN branch, we used the SE attention mechanism and depthwise separable convolution to focus on extracting local detailed features of the HSI as supplementary features. Based on the dual-branch multi-fusion network, rich spatial–spectral features can be fully extracted at multiple scales and extensively learned. Our experiments on three benchmark data sets showed that the suggested DFGCN outperforms advanced algorithms in HSIC tasks in terms of classification results.

The recently proposed SPSM (Superpixel–Subpixel Multilevel Network) is a three-branch network that reduces information loss by compensating for defects at different levels. We are considering conducting multi-branch explorations based on the DFGCN in the future.

## Figures and Tables

**Figure 1 sensors-24-04760-f001:**
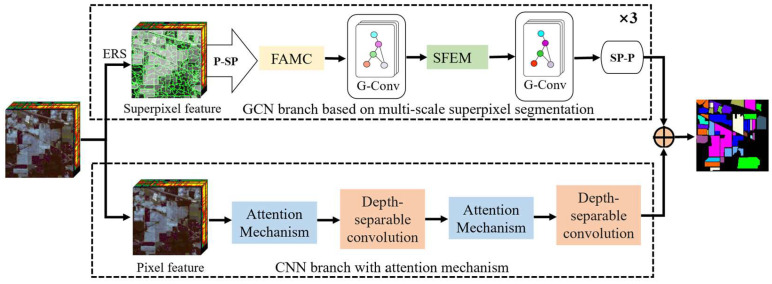
An outline of the proposed DFGCN for HSIC. It consists of two branches: a GCN branch, based on multi-scale superpixel segmentation, and a CNN branch with an attention mechanism.

**Figure 2 sensors-24-04760-f002:**
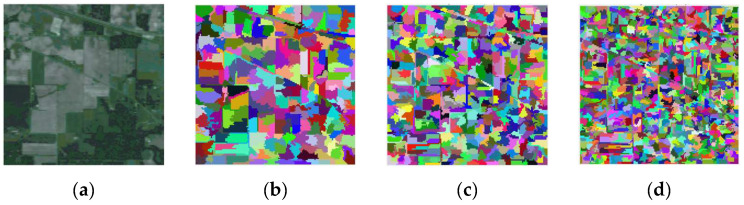
Segmentation maps acquired from the Indian Pines data set using the first principal component (PC) and adaptive multi-scale superpixel segmentation. Superpixel numbers at varying scales make up the figures: (**a**) first PC, (**b**) 262, (**c**) 525, and (**d**) 1051.

**Figure 3 sensors-24-04760-f003:**
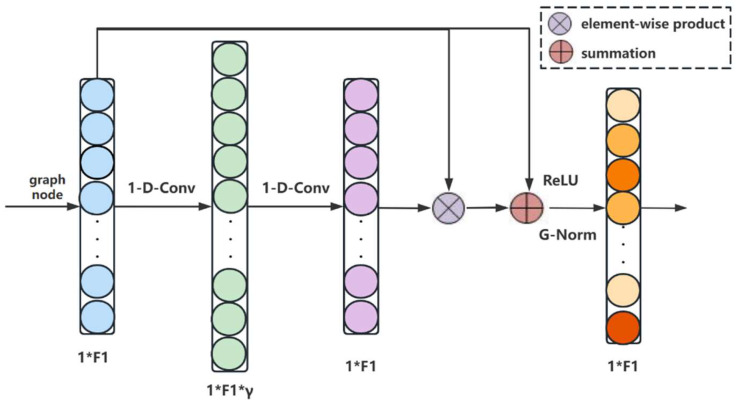
Implementation of the proposed spectral feature enhancement module.

**Figure 4 sensors-24-04760-f004:**
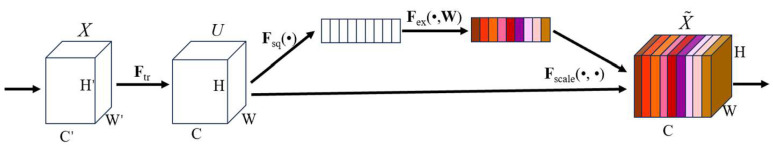
Structural diagram of SE attention mechanism.

**Figure 5 sensors-24-04760-f005:**
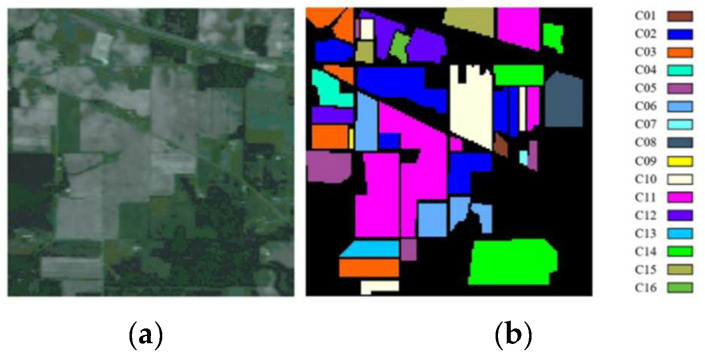
Indian Pines: (**a**) false-color synthetic image; (**b**) ground truth.

**Figure 6 sensors-24-04760-f006:**
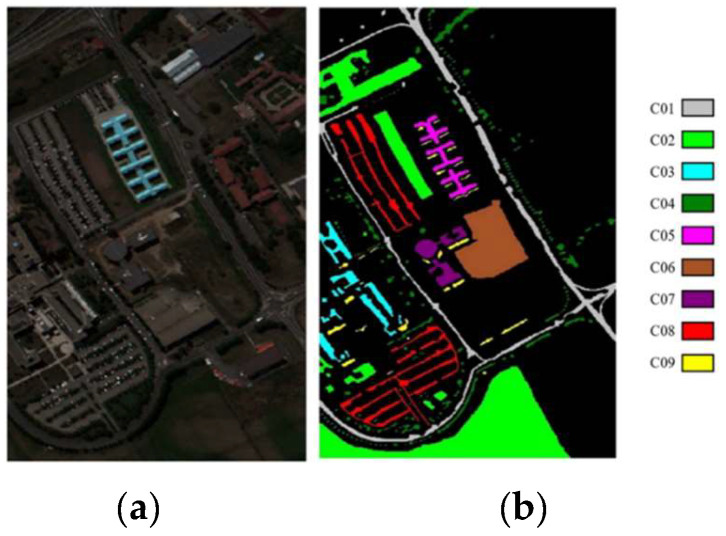
Pavia University: (**a**) false-color synthetic image; (**b**) ground truth.

**Figure 7 sensors-24-04760-f007:**
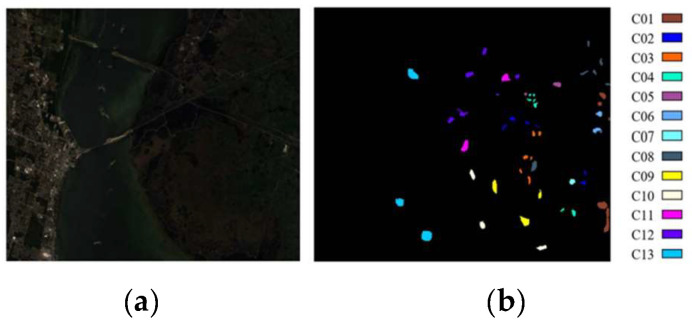
Kennedy Space Center: (**a**) false-color synthetic image; (**b**) ground truth.

**Figure 8 sensors-24-04760-f008:**
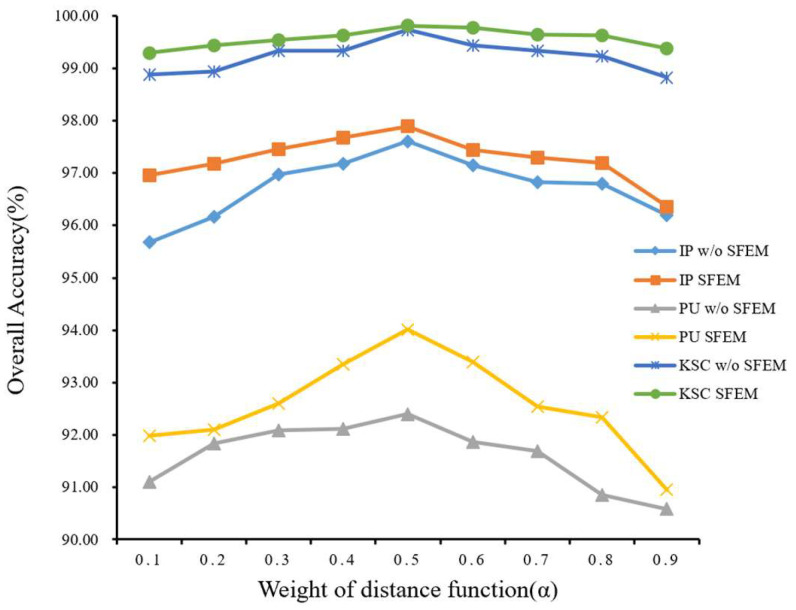
Impact of the parameters of α and the spectral feature enhancement module.

**Figure 9 sensors-24-04760-f009:**
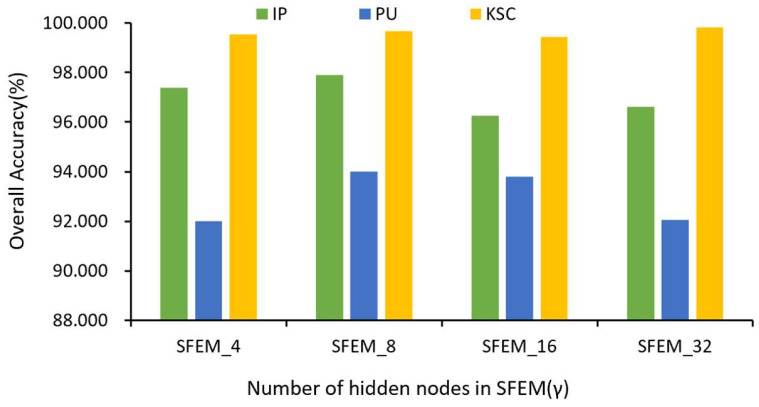
Performance of classification with varying values of γ in spectral feature enhancement.

**Figure 10 sensors-24-04760-f010:**
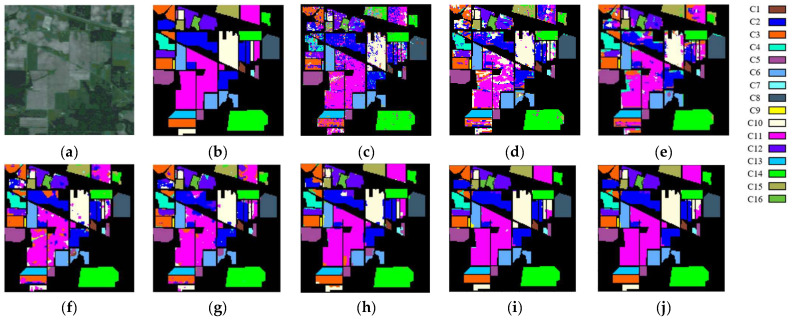
Maps of classification produced using the IndianP data set using various methods in an optimal close-up view: (**a**) original image; (**b**) ground truth; (**c**) RBF-SVM; (**d**) 2-D-CNN; (**e**) 3-D-CNN; (**f**) GCN; (**g**) S^2^GCN; (**h**) MDGCN; (**i**) SGML; and (**j**) DFGCN.

**Figure 11 sensors-24-04760-f011:**
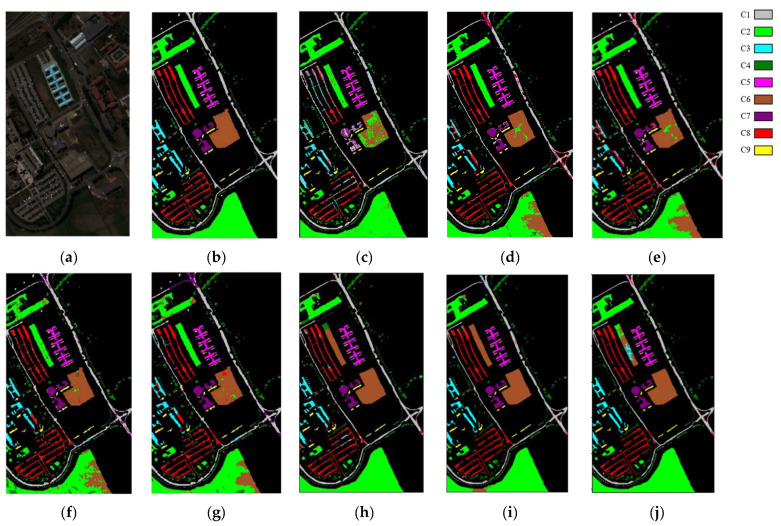
Maps of classification produced using the PaviaU data set using various methods in an optimal close-up view: (**a**) original image; (**b**) ground truth; (**c**) RBF-SVM; (**d**) 2-D-CNN; (**e**) 3-D-CNN; (**f**) GCN; (**g**) S^2^GCN; (**h**) MDGCN; (**i**) SGML; and (**j**) DFGCN.

**Figure 12 sensors-24-04760-f012:**
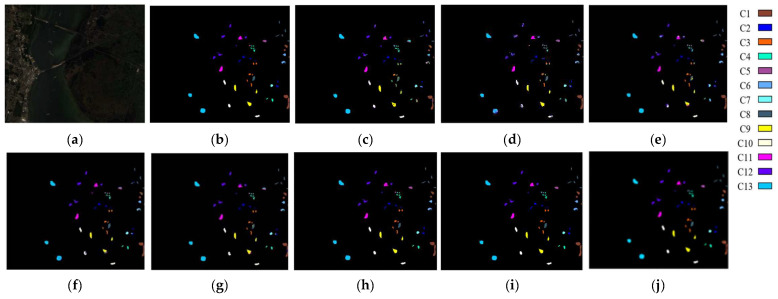
Maps of classification produced using the KSC data set using various methods in an optimal close-up view: (**a**) original image; (**b**) ground truth; (**c**) RBF-SVM; (**d**) 2-D-CNN; (**e**) 3-D-CNN; (**f**) GCN; (**g**) S^2^GCN; (**h**) MDGCN; (**i**) SGML; and (**j**) DFGCN.

**Figure 13 sensors-24-04760-f013:**
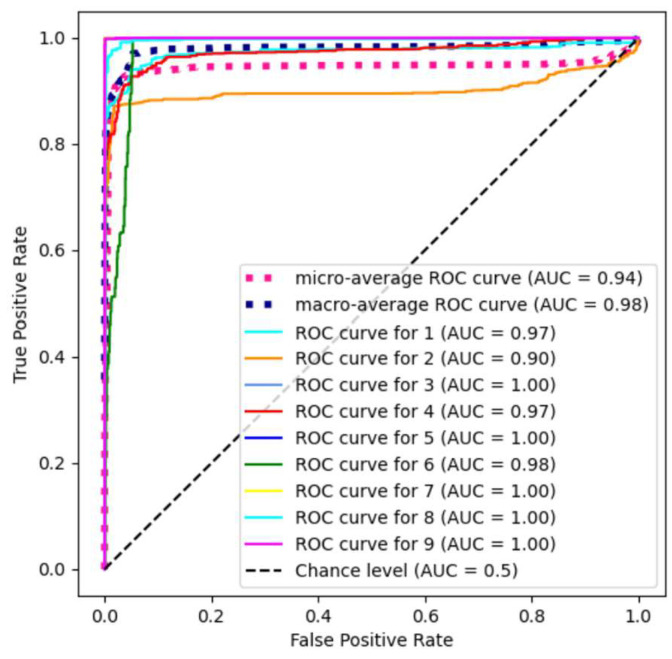
ROC curves and AUC values of each category in the PU data set.

**Figure 14 sensors-24-04760-f014:**
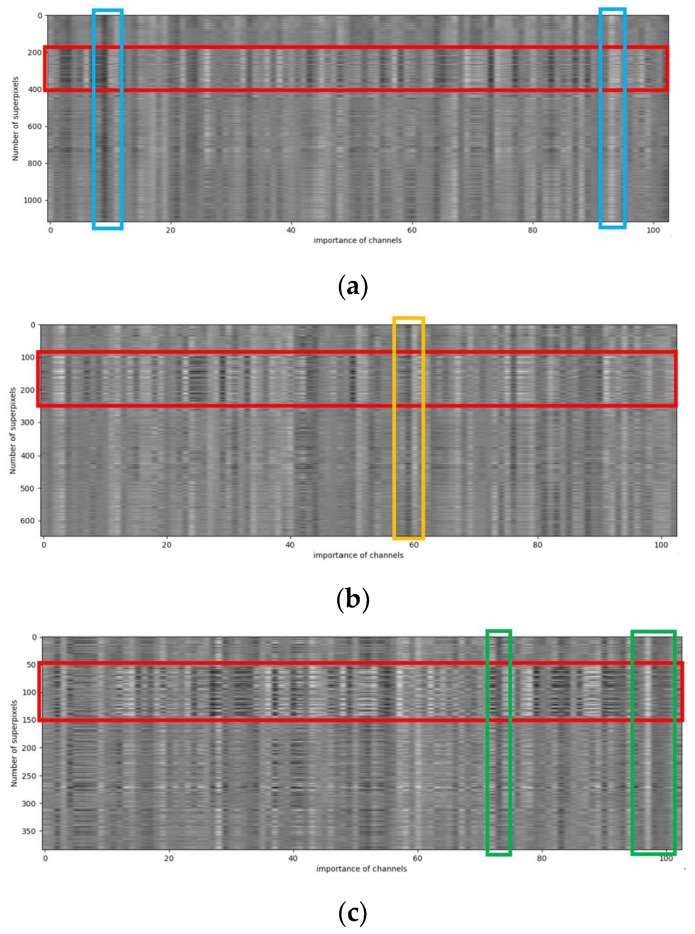
Multi-scale channel importance visualization using the PaviaU data set. (**a**–**c**) are weight visualizations of different scales. The red boxes indicate that the importance of superpixels within these ranges remains highly consistent across all channels at different scales, while the blue, orange, and green boxes represent that the importance of all superpixels on channels within these ranges is almost the same.

**Figure 15 sensors-24-04760-f015:**
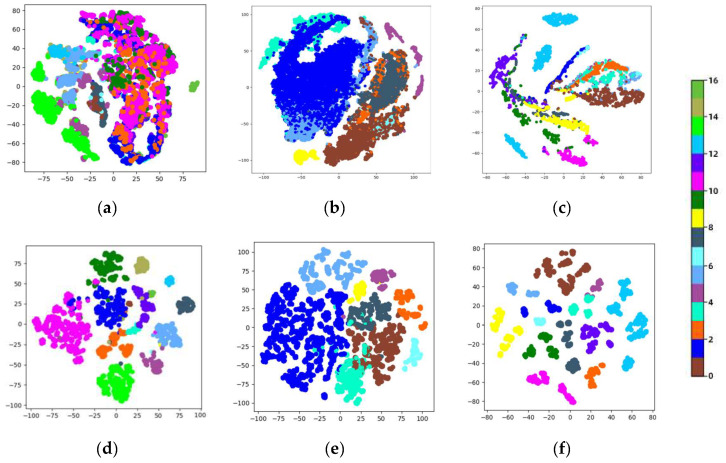
The visualization of features from the IndianP, PaviaU, and KSC data sets using 2-D t-SNE. (**a**–**c**) are the original feature spaces of labeled samples and (**d**–**f**) are the data distributions of labeled samples in the graph convolution feature space. Classes are represented by different colors.

**Table 1 sensors-24-04760-t001:** The training, test, and total sample numbers and types of labeled land covers in the IndianP data set.

ID	Land Cover Type	Train	Test	Total
01	Alfalfa	15	39	54
02	Corn—no-till	50	1384	1434
03	Corn—min-till	50	784	834
04	Corn	50	184	234
05	Grass—pasture	50	447	497
06	Grass–trees	50	697	747
07	Grass—pasture—mowed	15	11	26
08	Hay—windrowed	50	439	489
09	Oats	15	5	20
10	Soybean—no-till	50	918	968
11	Soybean—min-till	50	2418	2468
12	Soybean—clean	50	564	614
13	Wheat	50	162	212
14	Woods	50	1244	1294
15	Buildings–Grass–Trees–Drives	50	330	380
16	Stone–Steel–Towers	50	45	95
Total		695	9761	10,366

**Table 2 sensors-24-04760-t002:** The training, test, and total sample numbers and types of labeled land covers in the PaviaU data set.

ID	Land Cover Type	Train	Test	Total
01	Asphalt	30	6822	6852
02	Meadows	30	18,656	18,686
03	Gravel	30	2177	2207
04	Trees	30	3406	3436
05	Painted metal sheets	30	1348	1378
06	Bare Soil	30	5074	5104
07	Bitumen	30	1326	1356
08	Self-Blocking Bricks	30	3848	3878
09	Shadows	30	996	1026
Total		270	43,653	43,923

**Table 3 sensors-24-04760-t003:** The training, test, and total sample numbers and types of labeled land covers in the KSC data set.

ID	Land Cover Type	Train	Test	Total
01	Scrub	30	731	761
02	Willow swamp	30	213	243
03	CP hammock	30	226	256
04	CP/Oak	30	222	252
05	Slash pine	30	131	161
06	Oak/Broadleaf	30	199	229
07	Hardwood swamp	30	75	105
08	Graminoid marsh	30	401	431
09	Spartina marsh	30	490	520
10	Cattail marsh	30	374	404
11	Salt marsh	30	389	419
12	Mudflats	30	473	503
13	Water	30	897	927
Total		390	4821	5211

**Table 4 sensors-24-04760-t004:** The outcome of different methods for the IndianP data set classification.

No	RBF-SVM	2-D-CNN	3-D-CNN	GCN	S^2^GCN	MDGCN	SGML	DFGCN
01	68.70	65.10	80.52	**100.00**	**100.00**	97.14	**100.00**	**100.00**
02	74.82	63.65	85.91	83.02	82.12	85.85	**94.08**	93.79
03	67.88	85.33	79.27	80.43	87.22	95.27	**97.58**	**97.58**
04	56.52	95.08	89.39	95.45	99.12	**100.00**	**100.00**	**100.00**
05	89.81	91.39	94.72	94.02	98.01	95.61	93.74	**97.99**
06	92.57	99.09	**99.80**	92.67	90.44	86.22	97.85	99.14
07	89.37	85.19	92.91	92.12	**100.00**	**100.00**	**100.00**	**100.00**
08	97.13	53.02	98.38	98.00	99.40	98.83	**100.00**	**100.00**
09	40.02	66.13	97.59	**100.00**	**100.00**	**100.00**	**100.00**	**100.00**
10	69.18	98.77	85.51	88.97	93.46	89.15	99.02	**99.46**
11	78.19	89.79	90.63	79.70	85.30	95.26	94.95	**97.35**
12	69.31	86.97	86.48	86.23	85.01	94.32	87.23	**96.45**
13	89.44	**100.00**	86.11	**100.00**	**100.00**	96.79	99.38	99.38
14	94.70	87.18	97.09	95.45	97.48	99.26	**99.84**	**99.84**
15	57.06	**100.00**	83.32	92.44	93.10	98.52	98.18	98.79
16	97.60	**100.00**	96.71	**100.00**	**100.00**	97.96	**100.00**	**100.00**
AA (%)	77.02	85.42	90.27	92.41	94.42	95.64	97.62	**98.74**
OA (%)	79.43	73.93	89.40	86.86	89.20	93.57	96.32	**97.77**
Kappa (%)	76.41	70.66	87.90	85.34	87.93	92.63	96.84	**97.44**
F1-Score (%)	78.29	74.37	86.07	85.34	88.20	89.32	91.37	**92.38**

**Table 5 sensors-24-04760-t005:** The outcome of different methods for the PaviaU data set classification.

No	RBF-SVM	2-D-CNN	3-D-CNN	GCN	S^2^GCN	MDGCN	SGML	DFGCN
01	81.61	78.22	94.40	84.28	83.34	**95.76**	90.43	91.35
02	88.18	79.33	**94.39**	77.30	81.98	86.93	86.32	90.88
03	35.33	66.61	74.62	84.12	79.56	92.75	99.86	**100.00**
04	69.38	**97.84**	96.57	93.98	97.70	87.84	87.21	89.35
05	98.23	99.82	**100.00**	99.66	**100.00**	98.71	99.70	99.92
06	66.90	93.39	86.07	94.12	96.03	**100.00**	99.98	99.98
07	41.11	79.31	89.62	96.00	97.98	98.15	**100.00**	**100.00**
08	70.29	96.70	88.71	95.23	95.00	93.02	98.38	**99.86**
09	99.81	95.60	99.82	96.20	**100.00**	93.57	99.90	99.89
AA (%)	72.32	87.42	91.58	91.21	92.40	94.08	95.75	**96.80**
OA (%)	79.21	83.89	85.37	85.12	86.97	91.56	91.45	**93.88**
Kappa (%)	71.83	79.13	88.70	78.97	83.30	89.04	88.93	**92.02**
F1-Score (%)	73.23	79.55	86.79	80.67	84.98	91.25	89.46	**93.53**

**Table 6 sensors-24-04760-t006:** The outcome of different methods for the KSC data set classification.

No	RBF-SVM	2-D-CNN	3-D-CNN	GCN	S^2^GCN	MDGCN	SGML	DFGCN
01	89.73	99.59	91.67	96.78	94.98	**100.00**	**100.00**	**100.00**
02	81.01	78.40	89.39	91.00	92.00	99.06	**100.00**	**100.00**
03	73.68	66.81	14.12	96.30	99.34	**100.00**	**100.00**	**100.00**
04	55.13	32.88	42.61	78.12	88.02	95.05	94.60	**96.85**
05	59.77	75.57	60.90	82.78	73.23	**100.00**	88.56	**100.00**
06	43.52	62.31	40.83	75.45	83.99	**100.00**	**100.00**	**100.00**
07	73.70	93.33	83.41	91.97	90.01	**100.00**	97.33	**100.00**
08	70.89	90.52	79.89	97.02	96.45	**100.00**	**100.00**	**100.00**
09	85.41	97.14	90.41	79.23	91.78	**100.00**	**100.00**	**100.00**
10	91.94	63.37	97.88	88.34	95.00	**100.00**	**100.00**	**100.00**
11	97.00	97.69	99.94	**100.00**	88.67	**100.00**	**100.00**	**100.00**
12	81.48	58.99	93.07	96.45	99.01	**100.00**	**100.00**	99.58
13	97.96	81.16	99.91	96.01	99.10	**100.00**	**100.00**	**100.00**
AA (%)	77.02	76.75	75.69	89.96	91.66	99.55	98.50	**99.72**
OA (%)	83.40	80.38	85.37	92.00	93.89	**99.73**	99.40	99.71
Kappa (%)	81.37	78.12	83.60	91.12	92.67	99.70	99.34	**99.79**
F1-Score (%)	78.37	79.12	84.54	91.12	93.66	98.85	99.12	**99.53**

**Table 7 sensors-24-04760-t007:** DFGCN ablation classification performance on IndianP, PaviaU, and KSC data sets.

Data	GCN Branch	CNN Branch	DFGCN
Indian	97.09	95.42	**97.77**
PaviaU	92.87	91.80	**93.88**
KSC	99.41	96.91	**99.71**

**Table 8 sensors-24-04760-t008:** Ablation results of SFEM on IndianP, PaviaU, and KSC data sets.

Data	SFEM ?*	AA ↑	OA ↑	Kappa ↑
Indian	×	97.97	97.08	96.66
	√	**98.74**	**97.77**	**97.44**
PaviaU	×	95.78	92.40	91.06
	√	**96.80**	**93.88**	**92.02**
KSC	×	99.41	99.37	99.43
	√	**99.72**	**99.71**	**99.79**

* The “?” here indicates whether this SFEM module has been added to the model. “×” and “√” respectively indicate whether the SFEM module has been added or not in this model. The meaning of “↑” is that after adding the SFEM module to the model, AA, OA, and Kappa have all been improved.

**Table 9 sensors-24-04760-t009:** The computational expense of training and testing of different techniques.

Methods	Time(s)	Indian	PaviaU	KSC
GCN	Training	443.78	935.45	52.34
	Test	0.03	**0.05**	**0.01**
S^2^GCN	Training	577.46	1215.67	63.87
	Test	0.03	0.11	0.02
MDGCN	Training	274.96	803.97	23.50
	Test	1.456	3.61	0.16
SGML	Training	4.61	9.96	15.96
	Test	0.15	0.31	0.13
DFGCN (ours)	Training	**21.68**	**144.44**	**11.00**
	Test	**0.02**	0.12	0.03

## Data Availability

All data sets are available at http://www.ehu.eus/ccwintco/index.php?title=Hyperspectral_Remote_Sensing_Scenes (accessed on 20 October 2022).

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
