# Peer review of "A Dual-Branch Fusion of a Graph Convolutional Network and a Convolutional Neural Network for Hyperspectral Image Classification"

_sensors, 2024, doi:10.3390/s24144760_

Round 1

Reviewer 1 Report

Comments and Suggestions for Authors

There are many areas that need improvement in the paper, and I will only point out some of the issues:

1.     Starting from the Second 2, the readability of the manuscript's textual expression and the corresponding figures and tables is poor.

2.     English abbreviations should be clearly explained when first appearing, Such as GCN, DFGCN, MFAM.

3.     Check the writing in the manuscript, eg. extra space on Line 180 ‘efficacy s’, inappropriate expression of (see Figure 1)in Line 149.

4.     Is the description of the objective function in formula (1) appropriate?

5.     Many thematic images only have scale annotations, but do not have rulers? Eg. Figs. 8 and 9.

6.     Line 508 t-SNE and Line 509 t-sne. It is necessary to unify the expression method to avoid similar errors

7.     The legends in the figures are incorrect, and the corresponding text expression is unclear, Eg Fig. 14

8.     In Fig. 10, is Ground truth method better than the authors’ method? If not, analyze in detail the reasons for the results. The same goes for other experimental results, the authors should not only focus on the scores obtained, but also pay attention to the actual results.

9.     Figure 13 and the corresponding text description make this section too obscure to read.

Comments on the Quality of English Language

Starting from the Second 2, the readability of the manuscript's textual expression and the corresponding figures and tables is poor.

Reviewer 2 Report

Comments and Suggestions for Authors

The authors are presenting a novel deep learning network for hyperspectral image classification. The usage of deep learning techniques for the classification of hyperspectral images is a growing field as manual labeling or standard methods like SVM are very often time consuming and reaching only moderate classifications results.

The authors present a good overview on the current state of research. They describe their newly developed approach in sufficient detail. Also the results, the discussion and the conclusion are presented in a well written way,

In the results I am missing some widely used assessment parameters that the authors could add like sensitivity, specificity, AUC and F1-Score. I would be great if the authors could add these (perhaps as a supplement) to make their results more comparable to other papers that use these parameters instead of only overall accuracy (OA), average accuracy (AA), and Kappa coefficient.

Round 2

Reviewer 1 Report

Comments and Suggestions for Authors

The paper has made significant improvements, but there are still aspects that need to be revised:

1. Avoid similar errors::Lines 292.

2. Line 361:“The false-color composite image and ground truth are displayed in Figure 7(a) and (b), respectively. can ben better as : “The false-color composite image and ground truth are displayed in Figure 7.”

3. The language expression still needs overall improvement.

4. Some figures lack legends.

5. Many variables are not italicized.

Comments on the Quality of English Language

The language expression still needs overall improvement.

Reviewer 2 Report

Comments and Suggestions for Authors

The authors present a new deep learning network for the classification of HSI datasets. They present a good overview on the current state of research and describe their new network architecture in sufficient detail. They compare their network architecture to other currently used networks. As they have added some of the assessment parameters I was missing I recommend to accept the manuscript for publication.
